# Liquid Core Detection and Strand Condition Monitoring in a Continuous Caster Using Optical Fiber

**DOI:** 10.3390/s22249816

**Published:** 2022-12-14

**Authors:** Deva Prasaad Neelakandan, Dinesh Reddy Alla, Jie Huang, Ronald J. O’Malley

**Affiliations:** 1Department of Material Science and Engineering, Missouri University of Science and Technology, Rolla, MO 65409, USA; 2Department of Electrical and Computer Engineering, Missouri University of Science and Technology, Rolla, MO 65409, USA

**Keywords:** liquid core, metallurgical length, strain-sensing, Fiber Bragg Grating, optical fiber, dynamic reduction, on-line monitoring

## Abstract

Real-time monitoring of the liquid core position during the continuous casting of steel has been demonstrated using low-cost distributed optical-fiber-based strain sensors. These sensors were installed on the containment roll support structures in the segments of a production continuous caster to detect the position of the solid–liquid interface and monitor the strand condition during the continuous casting. Distributed Fiber Bragg Grating sensors (FBGs) were used in this work to monitor strain at six roll positions in the caster. The sensor performance was first validated by comparing optical strain measurements with conventional strain gauge measurements in the lab. Next, optical strain measurements were performed on an isolated caster segment in a segment maintenance facility using hydraulic jacks to simulate the presence of a liquid core under the roll. Finally, the sensors were evaluated during caster operation. The sensors successfully detected the load increase associated with the presence of a liquid core under each instrumented roll location. Incidents of bulging and roll eccentricity were also detected using frequency analysis of the optical strain signal. The liquid core position measurements were compared using predictions from computer models (digital twins) in use at the production site. The measurements were in good agreement with the model predictions, with a few exceptions. Under certain transient caster operating conditions, such as spraying practice changes and SEN exchanges, the model predictions deviated slightly from the liquid core position determined from strain measurements.

## 1. Introduction

Modern continuous casting technology has many features to improve throughput, quality, and safe operation of the process [1,2,3,4,5,6]. On-line monitoring systems are available in plants to monitor operating variables such as spray practice, casting speed, mold copper temperatures, and cooling water temperatures [3,7,8]. Operating practices have been implemented based on knowledge gathered from plant measurements and mathematical modeling by examining factors that influence continuous caster operation, such as superheat, thermal conductivity of the solidifying steel shell, interfacial heat transfer in the mold, convection from cooling water, etc. [9]. These models are normally tuned to be conservative in nature, i.e., in a caster, the liquid core position in the strand is predicted conservatively to avoid catastrophic loss of containment of the liquid core [10,11].

There are several ways that the position of the liquid core has been determined in the past, such as through the use of load cells [11], strain gauges [12], load torque estimators [11], smart sensors [13], drive roll shimming [14], slab macro-etching [15], mathematical modeling [9], and so on. Interruption of production in a steel plant is undesirable and comes with substantial economic costs. Knowledge of the liquid core position in the caster allows the operator to properly adjust speed, cooling, and strand taper to optimize soft reduction to minimize centerline segregation in the cast section and to maximize casting speed to achieve high caster productivity while also avoiding costly breakout events [9].

Strain measurements on the support structures of continuous caster support rolls have been shown to be an effective method for detecting the presence of the liquid core under a roll [2] during casting. When liquid is present under a roll in the caster, additional ferrostatic pressure is imparted to the roll and the additional load is transferred to the roll support structure through intermediate support bearings. This additional load can be detected by measuring the small deflections in the support structure using strain gauges. When the liquid core moves under a roll, a sudden increase in strain is observed. The measurement principle has been demonstrated in previous plant trials by Gregurich et al. [3] and elsewhere, where conventional strain gauges were used to detect the liquid core presence at specific roll locations in the caster. 

The application of Fiber Bragg Gratings (FBGs) in structural health monitoring of civil engineering structures [16], industrial composite curing (such as concrete and glass/epoxy) [17,18], and rock bolt cavities in the mining industry [19] have been reported in recent years. Fiber optic sensors employing FBGs have proved [20] to be effective in measuring microstrain levels of strain while remaining insensitive to steam, water, and radio frequency (RF) noise, which are all common in the caster environment [21,22,23,24,25]. The ability to distribute multiple sensors along a single fiber makes FBG-based optical fiber sensing an ideal candidate to instrument a large number of rolls in a caster at low cost and with minimal invasiveness to the equipment to detect liquid core movement along the length of the entire caster.

In this paper, for the first time to our knowledge, the Fiber Bragg Gratings were employed for strain measurements in a steel plant by embedding the FBGs onto a continuous caster strand to monitor liquid core position during the continuous steel casting process. In the current work, the caster support beams at six roll positions in three segments of a continuous caster were instrumented with distributed optical fiber-based FBG sensors to measure the strain experienced when the liquid core passed the specific instrumented roll locations in the caster during normal caster operation as casting speed was varied. The resultant measured strains were used to identify the presence or absence of liquid in the core at these six roll locations during caster operation. The observed liquid core positions were compared with liquid core position predictions from the production sites’ on-line solidification models (digital twins). In addition, a more detailed analysis of the strain signals was conducted to identify anomalies in the caster operation related to shell bulging and roll warpage events during caster operation.

## 2. FBG Sensing Principle

The structure of a Fiber Bragg Grating (FBG) is shown in Figure 1. The FBG was formed by introducing a periodic variation in the refractive index in the core of the optical fiber. This periodic pattern functions as a notch filter, reflecting light with a particular wavelength and transmitting light at all other wavelengths. The periodic pattern can be created using different methods such as interference lithography, phase mask writing, and point-by-point writing [21,22,23].

The wavelength of the reflected light (λ_B_), termed the Bragg wavelength, is defined in Equation (1) [21,23,26]:(1)$$\lambda_{B}=2n_{\mathrm{eff}}\Lambda$$
where n_eff_ is the effective refractive index of the core of the optical fiber and Λ is the grating period.

When an FBG sensor is subjected to an applied strain or temperature change, the refractive index of the core of the optical fiber and the grating period will experience changes, which results in a shift of the Bragg wavelength. Assuming that an applied strain or temperature change to the FBG sensor is uniform, the Bragg wavelength shift Δλ_B_ introduced by a strain or a temperature change can be described in Equation (2) [21,23,26]:(2)$$\frac{\Delta \lambda_{B}}{\lambda_{B}}=\left(1-p_{e}\right)\Delta \varepsilon +\left(a_{\mathrm{cte}}+a_{\mathrm{cto}}\right)\Delta T$$
where p_e_, a_cte_, and a_cto_ are the strain-optic coefficient, thermal expansion coefficient, and thermos-optic coefficient, respectively. Therefore, by continuously monitoring the wavelength of reflected light, the Bragg wavelength shift can be determined, and the applied strain or temperature change experienced by the FBG sensor can be calculated.

For quasi-distributed measurement, the base frequency of each FBG can be uniquely selected and associated with a specific measurement location along the optical fiber. This allows many FBGs to be employed on a single fiber. 

## 3. Experimental Methods

The experimental program to demonstrate the suitability of employing fiber optic strain sensors for the detection of liquid core position in a continuous caster was conducted in three stages: The first stage employed sensor demonstration testing in the laboratory using a bend tester. The second stage of testing was conducted on an isolated caster segment off-line in a test stand at our industry partner’s segment repair shop. The final stage of testing was performed on an operating caster at our industry partner’s production site over a two-day period.

### 3.1. Laboratory Testing

A three-point bend test was performed in the lab to compare the strain measurements obtained using a commercially supplied optical FBG strain gauge (Aurora^TM^) [27], shown in Figure 2, and a conventional strain gauge (Omega^TM^) [28]. A rectangular bar was instrumented and loaded to the maximum capacity (80 kN) of the load frame at a rate of 700 N/s. Previous testing results [3] using strain gauges on the caster indicated that the measured strains on the roll support cross beam were on the order of 250 microstrain units in the presence of a liquid core. Tests were performed at similar microstrain levels to verify the measurement capability of the sensors.

An additional test was conducted in the lab with the FBG sensors to evaluate the method used to compensate the strain measurements for varying environment temperatures. To perform temperature compensation, a second strain gauge was applied adjacent and perpendicular to the primary strain gauge that was parallel to the bending direction to allow for temperature compensation, as shown in Figure 3. The configuration of the strain sensors on the test beam produced a positive strain response on the longitudinal (parallel) sensor and a negative strain response on the sensor that was transverse (perpendicular) during bending, but both sensors exhibited a positive response to temperature on heating, thus allowing the temperature and strain effects to be deconvoluted.

The temperature compensation concept was evaluated in the laboratory load frame using a chamber to heat the instrumented beam in the bend tester. The steel beam was heated to create a uniform temperature of 150 °C distributed over the entire block. A three-point bend test was then performed to check the sensor performance at varying strain and temperature.

### 3.2. Off-Line Testing in Segment Repair Facility

To evaluate the robustness of the FBG strain gauges on a caster segment and to demonstrate the ability to measure deflection strains under industrial conditions, additional testing was performed on an isolated caster segment in the industry partner’s segment repair shop. To simulate the loads during caster operation, hydraulic jacks were placed close to the split-roll bearings located near the center of the segment cavity, as shown in Figure 4b. A five-roll caster segment was instrumented with four FBG strain gauges, as shown in Figure 4c. The caster segment was held in position and clamped using hydraulic cylinders with a variable pressure pump on a segment test stand and load applied to the rolls using hydraulic jacks at different pressure levels to simulate the ferrostatic pressure of the liquid core.

### 3.3. On-Line Testing during Caster Operation

As noted previously, pairs of orthogonally oriented FBG sensors were installed at six roll locations in the caster at the industry partner’s facility during a 12 h maintenance outage, and data were collected over two days of caster operation. Supplemental caster operating data were also collected for the trial period, such as casting speed, tundish level, mold width, etc., as well as liquid core position predictions from two on-line caster models (digital twins). The instrumented locations were selected based on the expected liquid core position in the caster, as shown in Figure 5. 

Two sensors were placed on the top side of each roll support beam at each roll location, one parallel to the support beam and one perpendicular to the support beam, to measure strain and deconvolute the ambient temperature effects on the sensor measurement, as shown in Figure 6. The strain gauge attachment site was prepared by grinding the surface and then the adhesive was cured under a weighted block. The sensor assembly was then covered with a protective rubberized sheet, as shown in Figure 7.

## 4. Results and Discussion

### 4.1. Laboratory Measurement Results

The FBG sensor and strain gauge measurements from the laboratory three-point bend test were well correlated. However, the signal-to-noise ratio (SNR) for the FBG sensor was much higher than for the strain gauge. The estimated resolution of the strain gauge and the FBG sensor was 50 microstrain and 0.5 microstrain, respectively, as shown in Figure 8. The sensing range of the fiber optic sensor was shown to fall well within the sensing range required for the liquid core detection application.

The primary oriented sensor exhibited a positive change in strain with load, while the perpendicularly oriented sensor experienced a negative strain due to the Poisson effect during beam bending, as expected. The strain measured by the transverse sensor was approximately −12% of that measured by the strain sensor that was oriented in the direction of beam bending. However, the effect of temperature on the output of both FBG sensors was positive and comparable. As a result, subtraction of the transverse sensor signal output from the longitudinal sensor signal output can be performed to compensate for the temperature effect on the sensor outputs and isolate the strain measurements, as shown in Figure 9.

### 4.2. Results from Strain Measurement in an Off-Line Segment 

The FBG strain gauges were tested in an off-line segment to check its ability to withstand the environment and resistance to vibration. As the load was applied to the caster rolls using hydraulic jacks, the strain experienced was measured by the FBG strain gauge. A maximum load of 8000 psi was applied by the hydraulic jacks. The clamping cylinders applied a pressure of 2000 psi to the segment to hold the segment closed. The load was applied individually to each caster roll, as shown in Figure 10, and the variation in strain was recorded with increasing and decreasing loads. A sudden drop in strain level was observed when the load was removed. 

Further analysis of the strain data was conducted and the strain variation with load (psi) was determined. A near linear trend in strain was observed with increasing loads at each roll location, as shown in Figure 11.

The loads were also applied to the caster rolls by simultaneously placing a single hydraulic jack next to the roll bearing at Roll-1 as well as Roll-2, and the strain was measured with varying applied load, as shown in Figure 12. When the load was applied on one of the caster rolls, a minor level of strain was observed on adjacent unloaded caster roll. The strain crosstalk observed for varying loads is shown in Figure 13. The roll-to-roll crosstalk is small and was judged to be small enough not to interfere with liquid core detection.

The position of hydraulic jacks was also changed to the center of rolls between the bearings from positions adjacent to the bearing. The strain measured with varying loads was observed for this loading configuration in Figure 14. The results suggested that the sensors should also be sensitive to changes in slab width, enabling the sensors to be used across varying steelmaking practices and product widths.

The strain trend observed before and after the clamping pressure was applied is also shown in Figure 15. The application of clamping forces to the segment also created strain at the sensor located on the roll support beam. In all cases, the strain change from the application of load to the rolls was easily detected. However, the sensor drift observed during the day also highlighted the sensitivity of the FBG sensors to temperature. This observation prompted a change in sensor in the installation for on-line sensing on the caster to the two-sensor orthogonal configuration already discussed.

### 4.3. Plant Test Results

Strain measurements and caster operating data were collected over two days of operation. An example of the data collected in the trial is shown in Figure 16. Here, the raw microstrains on the two optical sensors at Roll-75 (transverse and longitudinal orientation, shown in Figure 16a,b) are shown for approximately an 18 h cast sequence of 20 heats. From the cast speed data, shown in Figure 16c, it can be seen that the cast speed was reduced from its target speed (approximately 46 inches per minute) to 20 inches per minute (ipm) at each ladle exchange, and the target operating cast speed for each heat was reduced late in the sequence as the slab width increased. These speed changes resulted in liquid core position movement in the caster. Figure 16d shows the net strain measured at Roll-75 for the entire sequence. A shift in strain is visible at each slowdown to 20 ipm from 45 ipm, but the strain shift disappears at casting speeds below 41 ipm, attributed to the absence of the liquid core under Roll-75 at this casting speed and lower.

The raw strain signals for all six sensor sets for this cast sequence are shown in Figure 17. In each case, the strain signal shows a rise and drop that is related to the temperature of the sensor in the casting environment plus the strain associated with roll loads during casting. The strains associated with roll load result in an increase in the longitudinal sensor strain. In contrast, the transverse strain decreases with increasing roll loads because of Poisson’s ratio during beam deflection. On the other hand, the temperature effect increases the strain on both sensor orientations equally. Subtraction of the transverse strain from the longitudinal strain removes the temperature effect and gives a net strain due to the roll loading. 

The net strains for all six roll locations for the 20-heat cast sequence are shown in Figure 18. These signals indicate the initial strain experienced by the caster during various events, such as at startup, at operating transients such as speed and width changes, at strand cap-off, and during steady-state operation. When combined with other caster data, such as model outputs for liquid core position prediction, the signal data can be used to calibrate and confirm the model predictions and detect anomalies in the process. 

An example of model predictions for liquid core position for two on-line continuous caster models employed in the test plant site is shown in Figure 19. If the predicted liquid core position is longer than the instrumented roll position, a high level of strain is expected to be observed on the sensor because of ferrostatic pressure. If the predicted liquid core position is shorter than the roll position, a lower strain level is expected at that roll location.

Figure 20, Figure 21 and Figure 22 show the liquid core predictions and the corresponding roll strains for the first 12 heats of a 20-heat cast sequence at Roll-64, -74, and -78. The correspondence of the strain level shifts with the movement of the liquid core position under each roll is evident. The correspondence suggests that the test plant’s caster models provide a reasonable indication of liquid core position at nominal steady-state caster operating conditions. The measurements also suggest that the models over predict the liquid core position to an extent. The absence of a load shift at 39,000 s in Figure 21 indicates an instance of over-prediction, where the liquid core is predicted to be present.

Figure 23 shows an expanded view of the raw microstrain signals at Roll-75 for a single heat. The variations (noise) in the signal were analyzed using FFT (Fast Fourier Transformation) analysis to compare the frequencies of the signal fluctuations to characteristic frequencies for the BH CC#1 caster based on the calculations in Table 1 for shell bulging and roll eccentricity. The FFT analysis is shown in Figure 24 for the eighth heat in the cast sequence highlighted in Figure 16. 

The FFT analysis of the transverse strain sensor signal from the eighth heat in the cast sequence identified two frequency peaks that appear to correspond with two distinct disturbance conditions in the caster.

At a steady-state casting speed of 44.7 ipm, a peak is observed in the disturbance frequency spectrum at 0.0234 Hz and a second peak at 0.0429 Hz. The peak at 0.0234 Hz matches closely with the calculated frequency for a bent or warped roll of 249–250 mm in diameter. Figure 24a shows that this matches rolls located either in Segment 1, 5, or 6.

The peak at 0.0429 Hz matches very closely with slab bulging in area 14 of the caster, which is the non-segmented part of the caster with a 436 mm roll pitch. Figure 24b shows these roll locations. For bulging to occur in this region, the liquid core must extend into the non-segmented part of the caster. The liquid core position model estimates from Figure 20 confirm that the liquid core does extend to Roll-83 under the casting conditions for heat eight, as shown in Figure 25.

The frequency analysis of the strain signals at Roll-75 identified two potential conditions that could affect slab quality: (1) a bent or warped roll condition in Segment 1, 5, or 6, and (2) a bulging condition in the non-segmented portion of the caster, resulting from running the liquid core beyond the segmented portion of the caster.

## 5. Conclusions

Optical-fiber-based sensors have been used successfully for the first time ever to measure the strains that affect the roll loads in the segments of a continuous caster in a working steel plant. The measurement technique can be employed for several possible objectives:The highly distributed optical strain measurement technique provides a low-cost method for continuously monitoring the casting process on a large number of rolls and segments for liquid core tracking and metallurgical length monitoring.The optical strain signals can also provide information about the strains experienced by the caster during various events, such as at startup, at operating transients such as speed and width changes, at strand cap-off, and during steady-state operation.In combination with other caster data, the strain signals can also be used to calibrate and validate continuous caster model predictions. These models can then be used with a higher degree of confidence to enhance control of the continuous casting process.FFT analysis of the strain data provides an on-line real-time system to monitor the condition of the rolls and the solidified strand during caster operation. The signals from the optical strain gauges have been shown to be capable of detecting the presence of bent or warped rolls and shell bulging during caster operation.

The viability of this optical fiber measurement method has been demonstrated in this short trial for identifying liquid core position movement for productivity tracking, soft reduction control, and model tuning along with other caster condition indicators that can be used to detect potential quality events, such as roll eccentricity and slab bulging. By monitoring the magnitude of these critical frequencies, critical thresholds can be set to identify internal quality issues before they become application-critical.

## Figures and Tables

**Figure 1 sensors-22-09816-f001:**
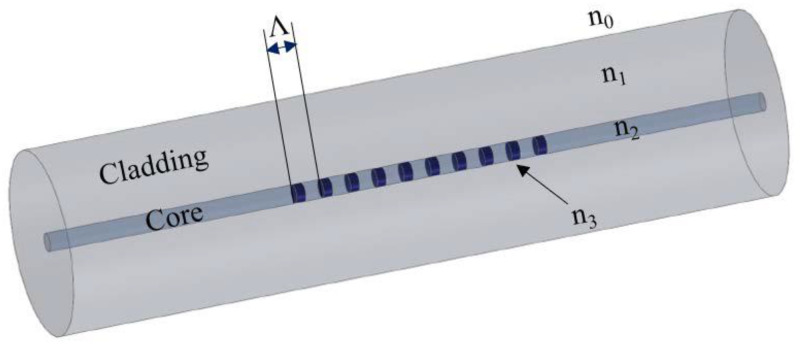
Schematic drawing of a Fiber Bragg Grating (FBG) structure in the core of an optical fiber.

**Figure 2 sensors-22-09816-f002:**
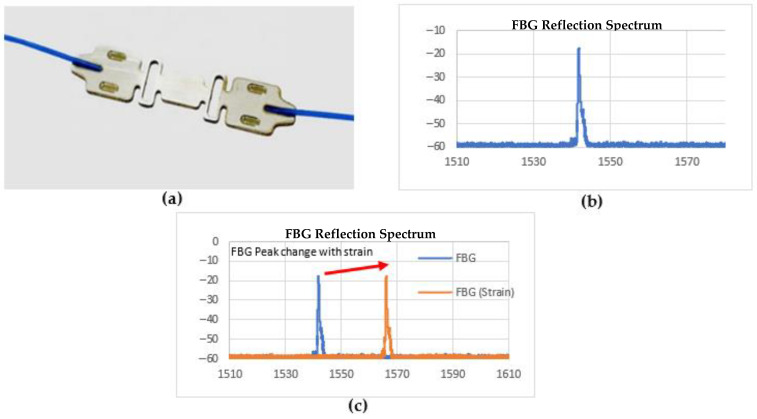
(**a**) Aurora^TM^ FBG integrated strain sensor, (**b**) FBG Reflection Spectrum, (**c**) FBG peak change with strain change.

**Figure 3 sensors-22-09816-f003:**
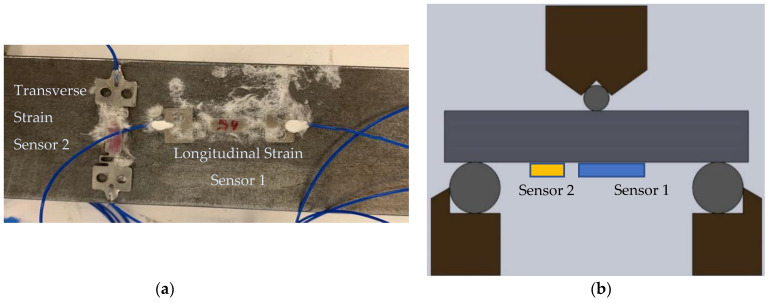
(**a**) FBG sensors 1 and 2 attached to steel block. (**b**) Schematic of a 3-point bend test setup with both sensors.

**Figure 4 sensors-22-09816-f004:**
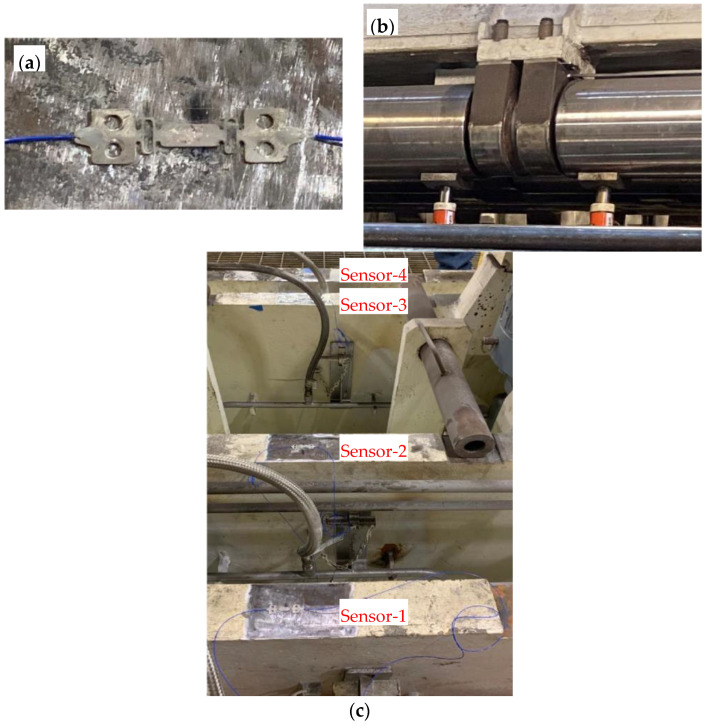
(**a**) FBG strain gauge superglued onto caster roll. (**b**) Hydraulic jacks placed close to bearings. (**c**) Four FBG strain gauges attached to four caster roll support beams.

**Figure 5 sensors-22-09816-f005:**
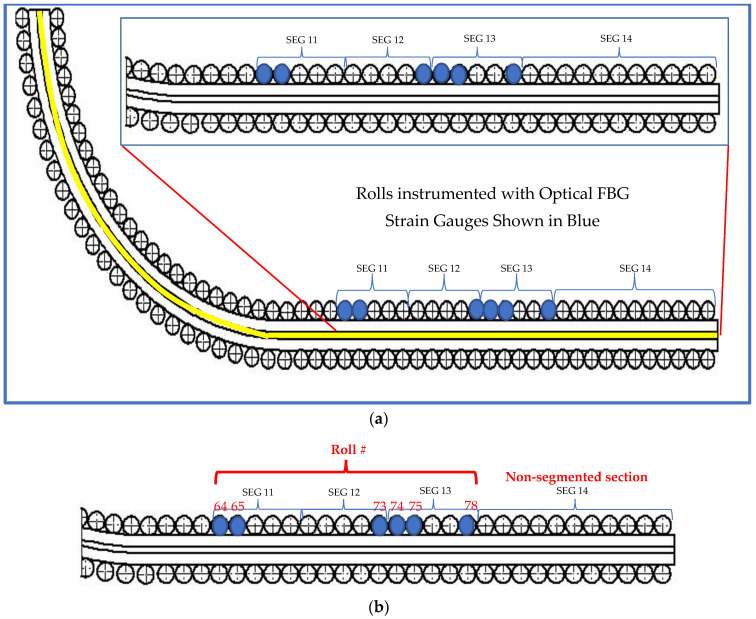
Location of rolls instrumented with fiber optic strain sensors, (**a**) Layout showing the caster segments and instrumented roll segments. The yellow line along the caster represents the region where liquid core is present for a specific operating condition. (**b**) Image shows roll location instrumented with optical FBG strain gauges and interrogated.

**Figure 6 sensors-22-09816-f006:**
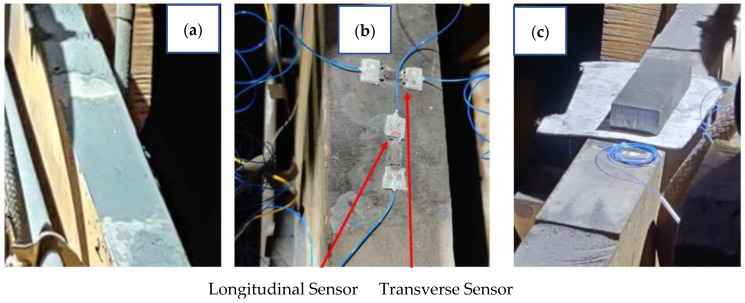
Installation of optical strain gauge sensors: (**a**) beam site preparation, (**b**) sensor placement, (**c**) curing of adhesive under a weight block.

**Figure 7 sensors-22-09816-f007:**
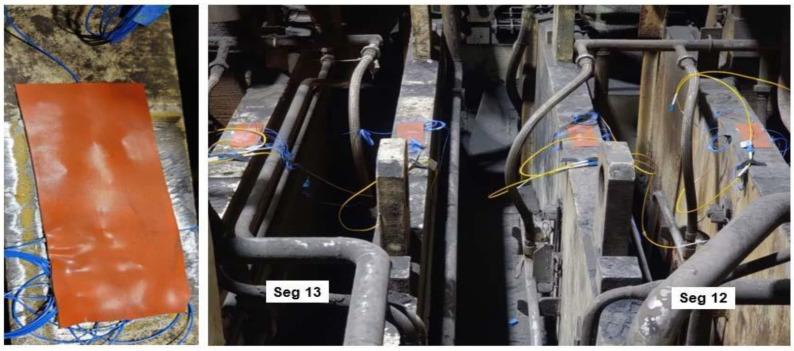
Final sensor installation with a rubberized protective covering.

**Figure 8 sensors-22-09816-f008:**
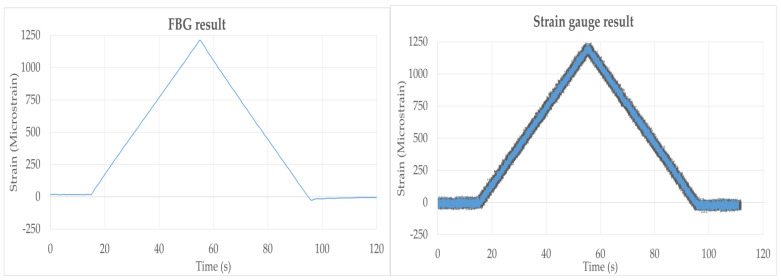
Strain signal output comparisons between FBG sensor and strain gauge.

**Figure 9 sensors-22-09816-f009:**
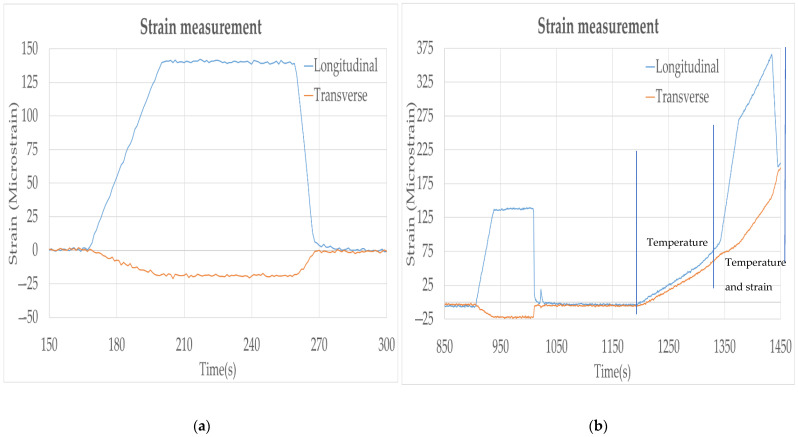
(**a**) Strain measured by longitudinal and transverse FBG sensors. (**b**) Strain measured by both sensors with varying temperature and strain.

**Figure 10 sensors-22-09816-f010:**
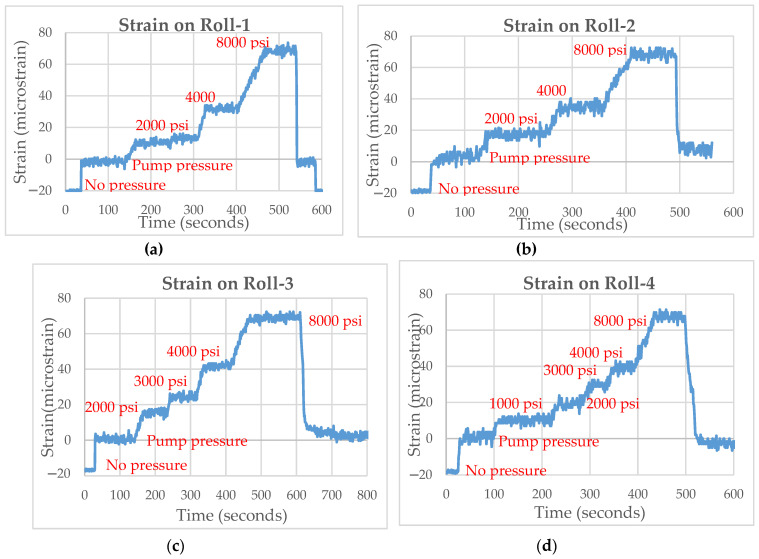
(**a**) Strains measured at Roll-1 for varying loads. (**b**) Strain measured by Roll-2 for varying loads, (**c**) Strain measured by Roll-3 for varying loads, (**d**) Strain measured by Roll-4 for varying loads.

**Figure 11 sensors-22-09816-f011:**
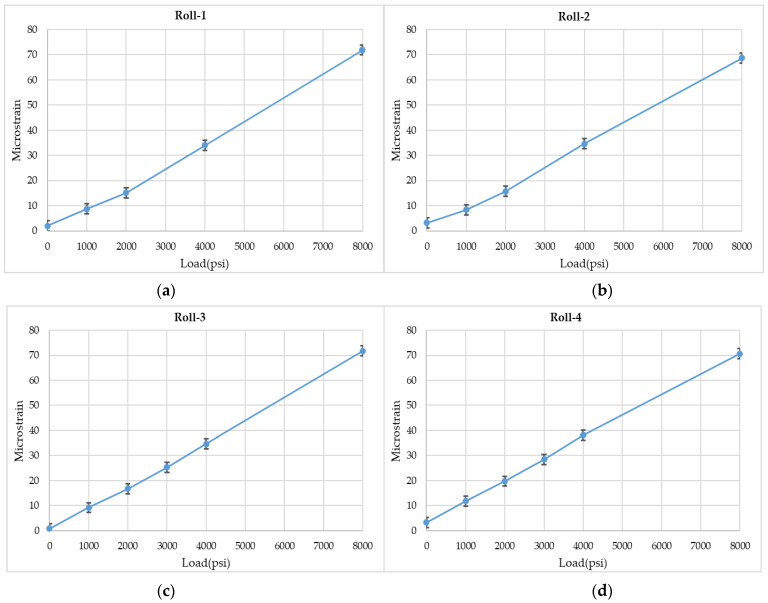
(**a**) Strain measured by Roll-1 for varying loads. (**b**) Strain measured by Roll-2 for varying loads. (**c**) Strain measured by Roll-3 for varying loads. (**d**) Strain measured by Roll-4 for varying loads.

**Figure 12 sensors-22-09816-f012:**
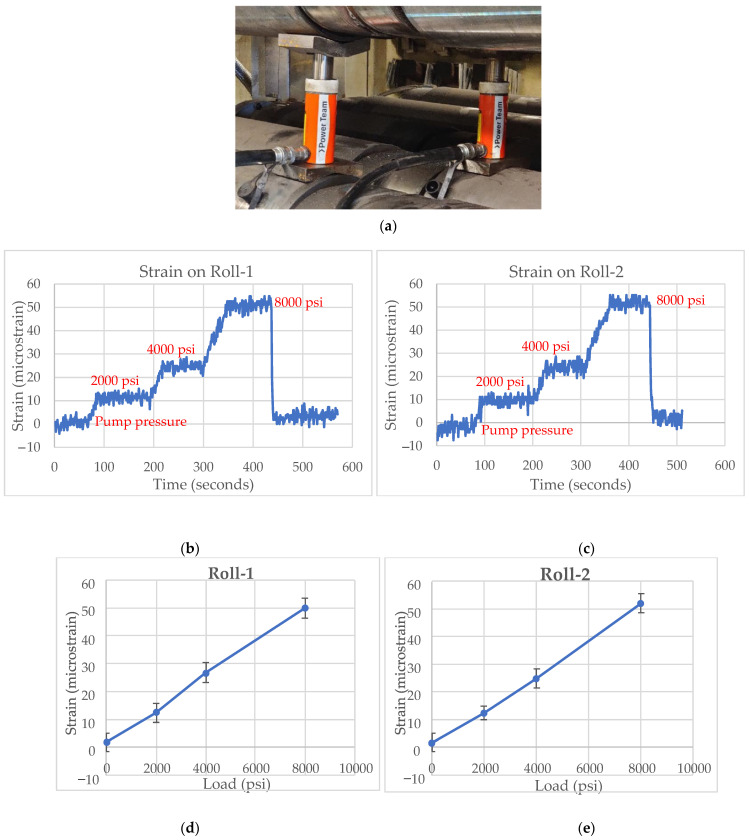
(**a**) Hydraulic jacks placed on Roll-1 and Roll-2. (**b**) Strain measured by Roll-1 for varying loads. (**c**) Strain measured by Roll-2 for varying load. (**d**) Strain vs. load (psi) on Roll-1. (**e**) Strain vs. load (psi) on Roll-2.

**Figure 13 sensors-22-09816-f013:**
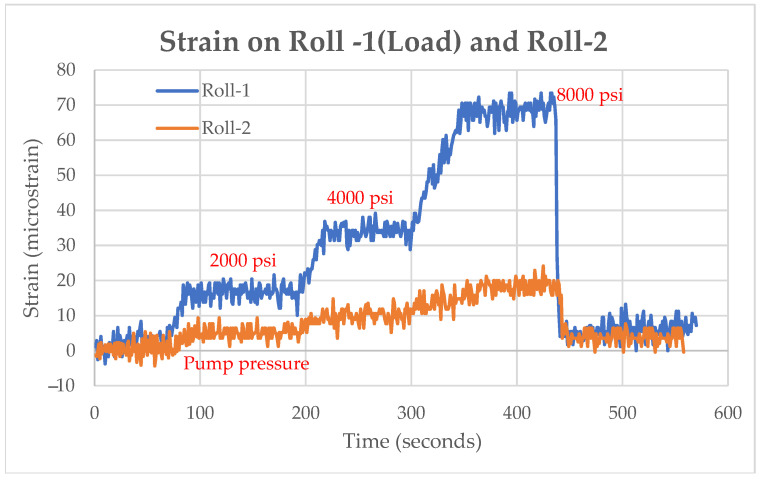
Strain crosstalk observed on Roll-2 when load is applied to Roll-1.

**Figure 14 sensors-22-09816-f014:**
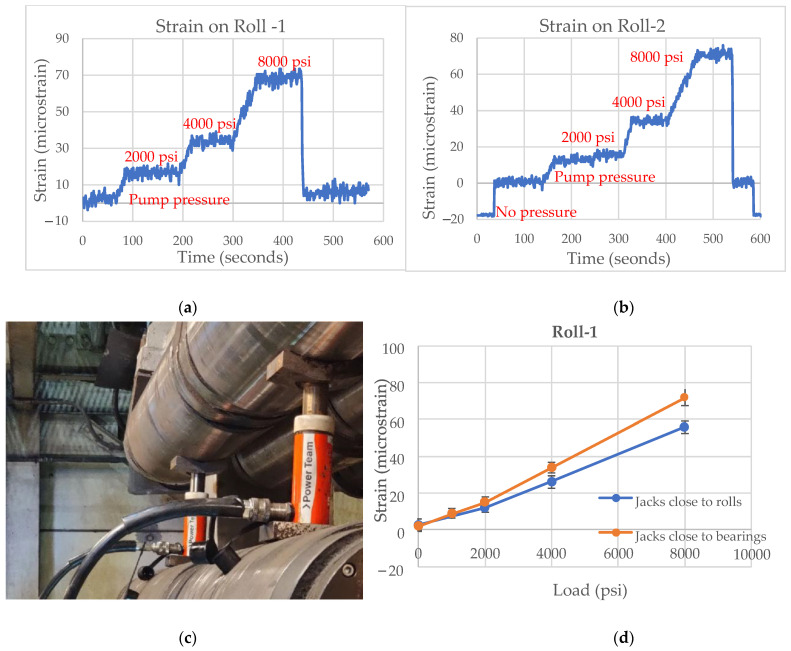
(**a**) Strain measured by Roll-1 with single hydraulic jack. (**b**) Strain measured by Roll-2 with single hydraulic jack. (**c**) Hydraulic jacks placed close to rolls. (**d**) Strain with respect to load (psi) on Roll-1.

**Figure 15 sensors-22-09816-f015:**
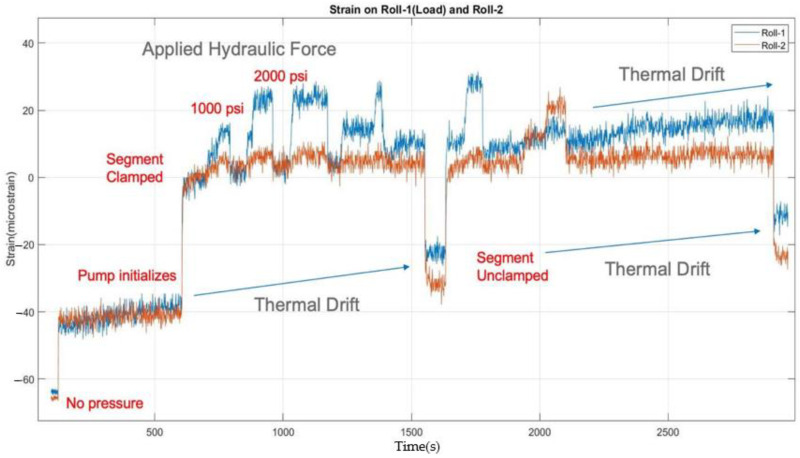
Strain measured with varying segment clamping pressures and hydraulic jack loads.

**Figure 16 sensors-22-09816-f016:**
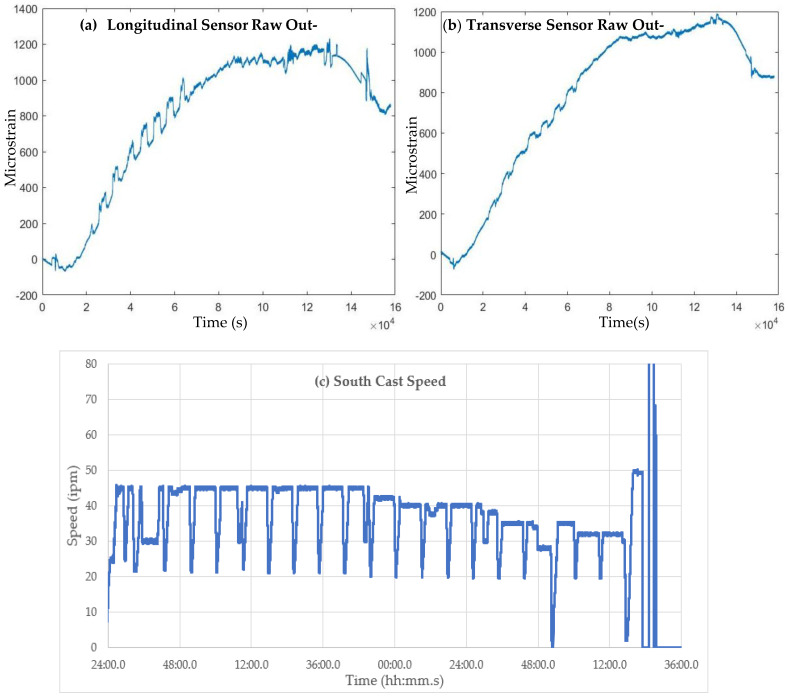
Raw optical sensor microstrain measurements for a 20-heat cast sequence: (**a**) longitudinal sensor output, (**b**) transverse sensor output, (**c**) casting speed for sequence, and (**d**) net temperature compensated strain (signal a–signal b).

**Figure 17 sensors-22-09816-f017:**
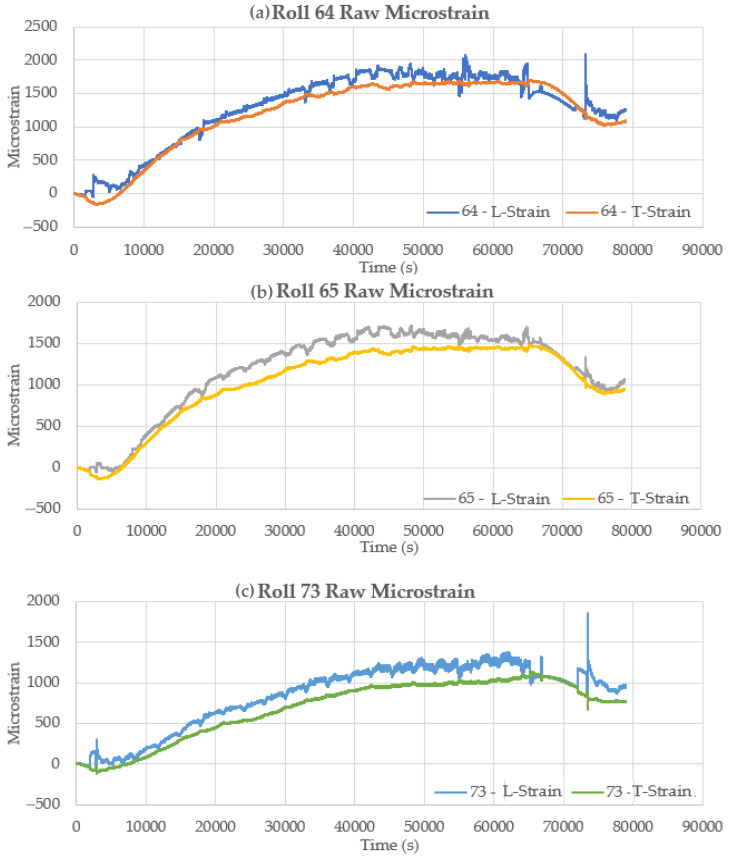
Raw longitudinal and transverse strain signals for rolls (**a**) 64, (**b**) 65, (**c**) 73, (**d**) 74, (**e**) 75, and (**f**) 78.

**Figure 18 sensors-22-09816-f018:**
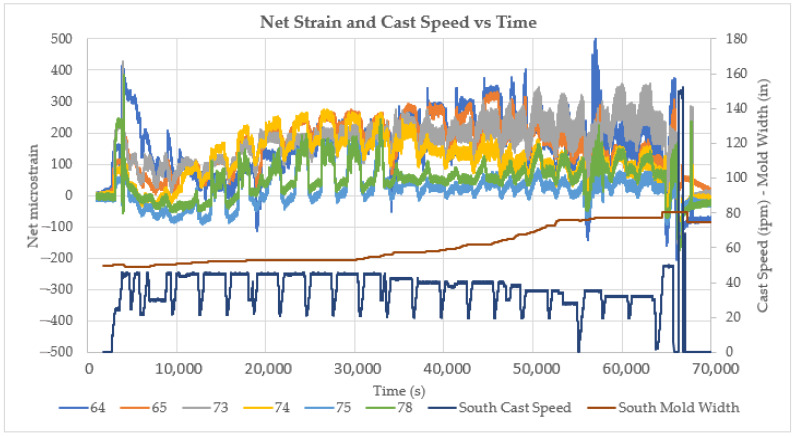
Net strain at all roll locations superimposed on casting speed and slab width data for a 20-heat cast sequence over approximately 14 h of caster operation.

**Figure 19 sensors-22-09816-f019:**
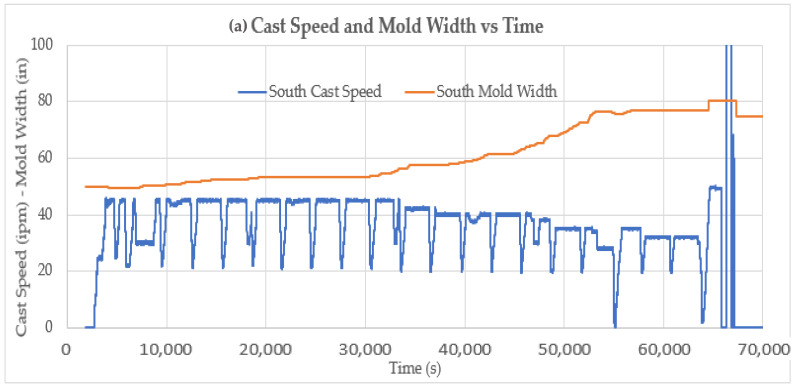
(**a**) Cast speed and mold width vs. time. (**b**) Cast speed, width, and liquid core position predictions from two models with instrumented roll positions superimposed for a 20-heat cast sequence.

**Figure 20 sensors-22-09816-f020:**
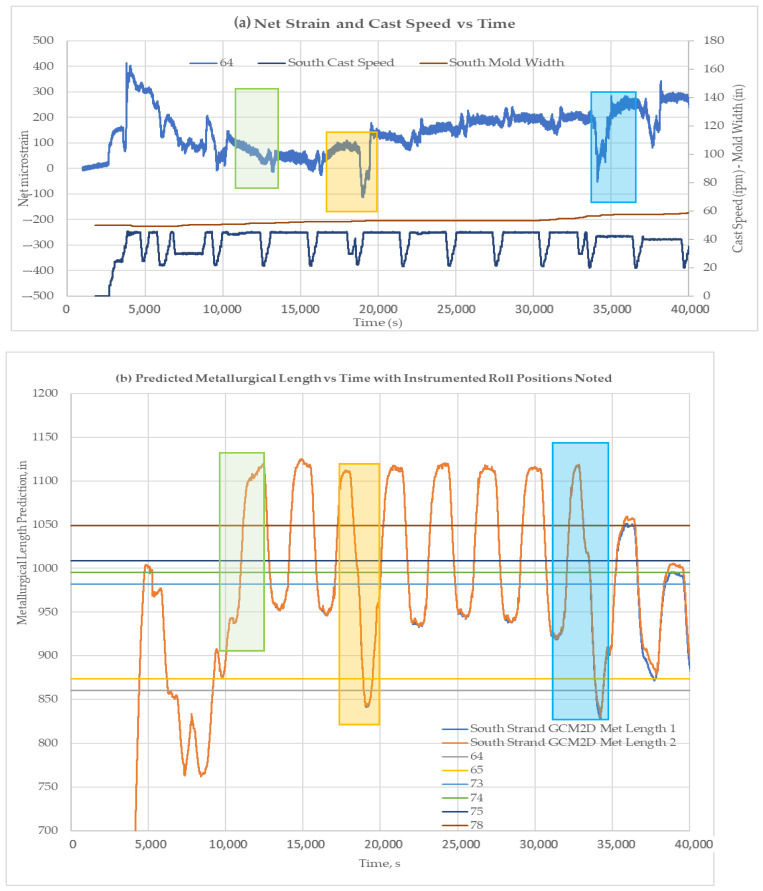
Liquid core position and corresponding net roll strains for the first 12 heats of the 20-heat cast sequence with liquid core locations highlighted for Roll-64 (all roll strains shown). (**a**) Net strain and cast speed is plotted against time. Three sections (highlighted by rectangles) were chosen to be compared with metallurgical length. (**b**) Predicted metallurgical lengths for different time periods. Strain comparison for sections highlighted in Figure 21a with metallurgical lengths.

**Figure 21 sensors-22-09816-f021:**
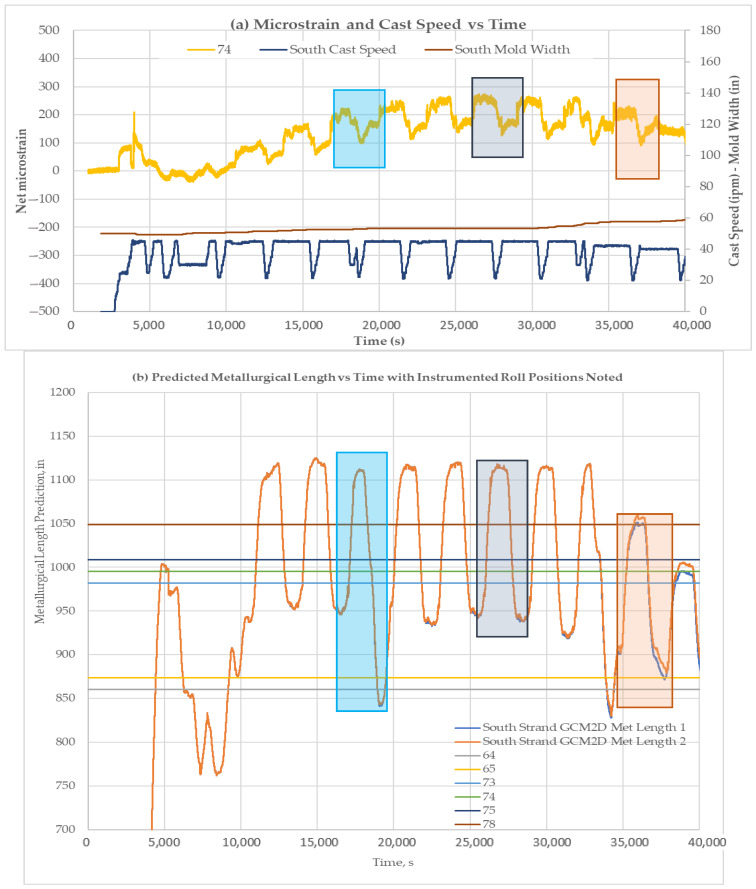
Liquid core position and corresponding net roll strains for the first 12 heats of the 20-heat cast sequence with liquid core locations highlighted for Roll-74. (**a**) Microstrain and cast speed is plotted against time. Three sections (highlighted by rectangles) were chosen to be compared with metallurgical length. (**b**) Predicted metallurgical lengths for different time periods. Strain comparison for sections highlighted in Figure 22a with metallurgical lengths.

**Figure 22 sensors-22-09816-f022:**
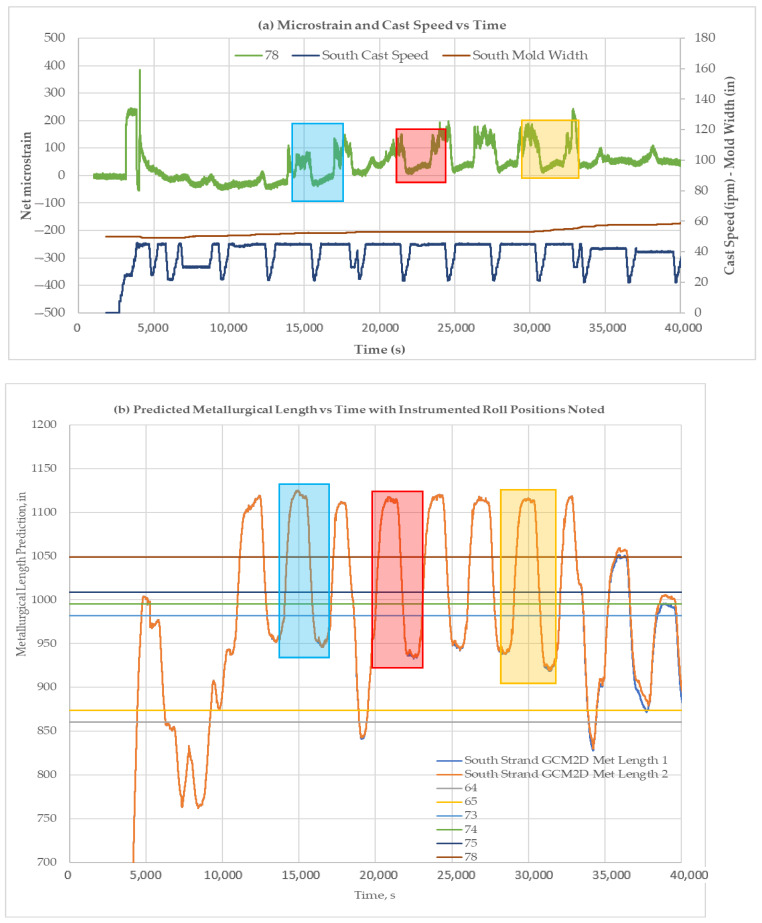
Liquid core position and corresponding net roll strains for the first 12 heats of the 20-heat cast sequence with liquid core locations highlighted for Roll-78. (**a**) Microstrain and cast speed is plotted against time. Three sections (highlighted by rectangles) were chosen to be compared with metallurgical length. (**b**) Predicted metallurgical lengths for different time periods. Strain comparison for sections highlighted in Figure 23a with metallurgical lengths.

**Figure 23 sensors-22-09816-f023:**
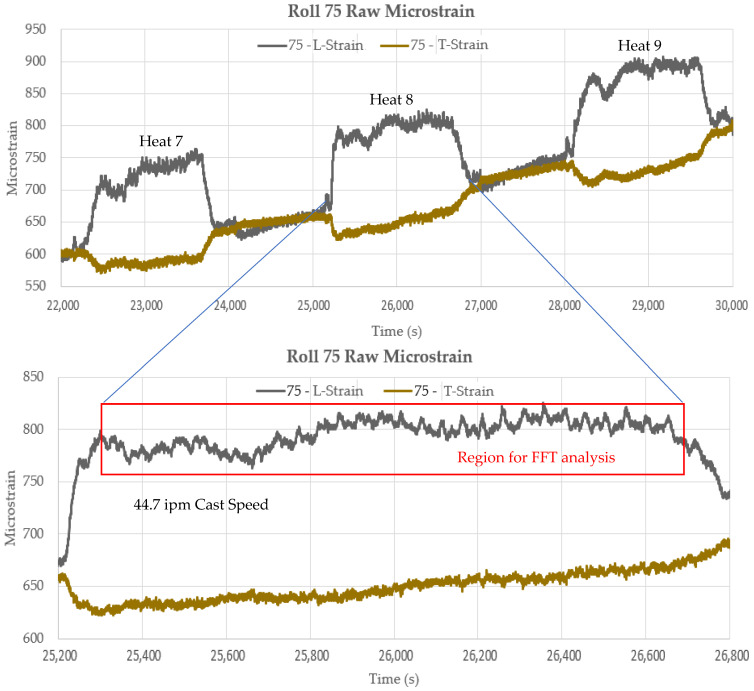
Typical raw strain signal fluctuations in presence of liquid core loading.

**Figure 24 sensors-22-09816-f024:**
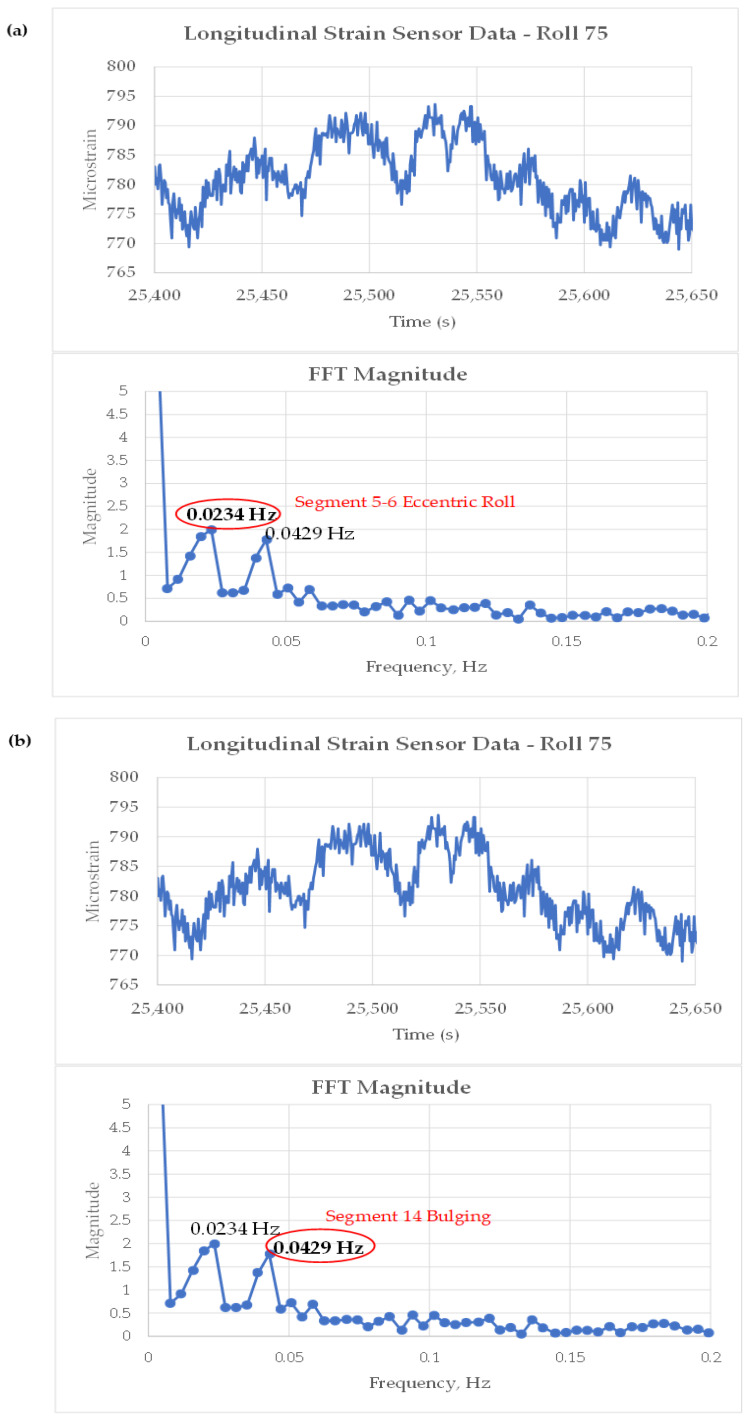
FFT analysis of longitudinal strain sensor signals from heat 8 in the cast sequence and highlighted locations for possible locations in the caster that might initiate the event based on matched frequency. (**a**) Chart representing roll eccentricity frequency in Segments 5–6. (**b**) Chart representing bulging frequency in non-segmented section 14. (**c**) Table representing eccentricity frequency for Segments 5–6. (**d**) Highlighted region in Table representing bulging frequency for Segment 14.

**Figure 25 sensors-22-09816-f025:**
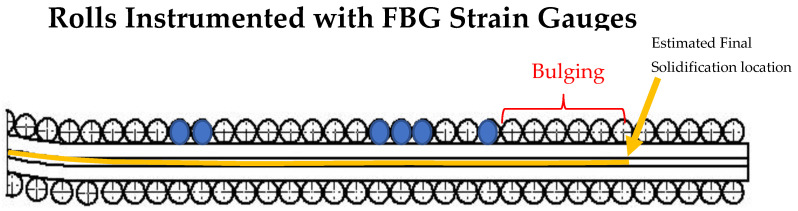
Liquid core position predicted by Burns Harbor caster model for heat 8.

**Table 1 sensors-22-09816-t001:** Characteristic Disturbance Frequencies Calculated for Roll Eccentricity and Inter-Roll Bulging.

Characteristic Frequencies for ArcelorMittal BH CC1											
Segment	Roll	Diameter	Pitch	Roll Diameter Frequency (Hz) at Speed (ipm)				Roll Pitch Frequency (Hz) at Speed (ipm)			
				30	35	40	45	30	35	40	45
FR	1	134.6	110.4	0.030	0.035	0.040	0.045	0.115	0.134	0.153	0.173
0	2	134.6	178	0.030	0.035	0.040	0.045	0.071	0.083	0.095	0.107
0	3	134.6	190	0.030	0.035	0.040	0.045	0.067	0.078	0.089	0.100
0	4	134.6	204	0.030	0.035	0.040	0.045	0.062	0.073	0.083	0.093
0	5	134.6	226	0.030	0.035	0.040	0.045	0.056	0.066	0.075	0.084
0	6	215.9	247	0.019	0.022	0.025	0.028	0.051	0.060	0.069	0.077
0	7	215.9	253	0.019	0.022	0.025	0.028	0.050	0.059	0.067	0.075
0	8	215.9	245	0.019	0.022	0.025	0.028	0.052	0.060	0.069	0.078
0	9	215.9	249	0.019	0.022	0.025	0.028	0.051	0.060	0.068	0.077
1	10–14	248.9	300	0.016	0.019	0.022	0.024	0.042	0.049	0.056	0.064
2	15–19	279.4	224	0.014	0.017	0.019	0.022	0.057	0.066	0.076	0.085
3	20–24	330.2	376	0.012	0.014	0.016	0.018	0.034	0.039	0.045	0.051
4	25–29	330.2	393	0.012	0.014	0.016	0.018	0.032	0.038	0.043	0.048
5	30–36	250	368	0.016	0.019	0.022	0.024	0.035	0.040	0.046	0.052
6	37–43	250	312	0.016	0.019	0.022	0.024	0.041	0.047	0.054	0.061
7	44–48	300	341	0.013	0.016	0.018	0.020	0.037	0.043	0.050	0.056
8	49–53	300	358	0.013	0.016	0.018	0.020	0.035	0.041	0.047	0.053
9	54–58	300	355	0.013	0.016	0.018	0.020	0.036	0.042	0.048	0.054
10	59–63	300	350	0.013	0.016	0.018	0.020	0.036	0.042	0.048	0.054
11	64–68	300	347	0.013	0.016	0.018	0.020	0.037	0.043	0.049	0.055
12	69–73	300	340	0.013	0.016	0.018	0.020	0.037	0.044	0.050	0.056
13	74–78	300	340	0.013	0.016	0.018	0.020	0.037	0.044	0.050	0.056
14	79–89	482.6	346	0.008	0.010	0.011	0.013	0.037	0.043	0.049	0.055

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
