# Peer review of "Liquid Core Detection and Strand Condition Monitoring in a Continuous Caster Using Optical Fiber"

_sensors, 2022, doi:10.3390/s22249816_

Round 1

Reviewer 1 Report

The manuscript reports on an installation of commercially available fiber Bragg gratings in a casting plant for condition monitoring and liquid-core detection. During caster operation, strain measurements were carried out on roll support structures and the conditions of the caster is deduced from these strain measurements. The authors refer to similar investigations that were carried out with strain gauges. According to the authors, the advantages of using the FBG instead of strain gauges are the multiplexing capability, the immunity to EMI and the lower costs of the FBG.

I do not recommend this manuscript for publication in Sensors as it describes a use of commercial FBG sensors on support structures and thus does not describe any novelty content with regard to sensor technology. Furthermore, the manuscript contains considerable weaknesses that argue against publication. Some examples of these are

·       Figure 9a and 9b show FBG signals without temperature compensation. Figure 9c should depict the temperature-compensated FBG signals. The time scales between Figures 9a, 9b, and 9c don’t match to each other. Therefore, you can’t deduce from Figure 9c how well the temperature compensation works. What is finally shown in Figure 9c? Was there no strain on the specimen but only temperature changes and we see the limits how well the temperature changes worked?

·        Figure 10 has no labelling at all on its horizontal axis

·        The experiment with the off-line segment was performed without the transversal strain gages and were therefore invalid. Why are thus these experiments reported at all?

Author Response

PFA the response

Reviewer 2 Report

I have read the paper carefully and found it an interesting work. The introduction part is supported by relevant papers in literature. The sections in the paper are well organised. Results are clearly presented. The language used in the paper is acceptable. The results presented in the paper are supported by well explained graphs and the scientific soundness of the results given in the paper is high. In my view, the paper deserves publication in a good journal like sensors. I recommend to publish this paper. 

Author Response

PFA the response

Author Response

PFA the response

Round 2

Reviewer 1 Report

Based on the authors' responses, I recognise the use and liquid core detection of FBG sensors in the harsh circumstances of a casting process as a novelty.

Reviewer 3 Report

Accept in present form